# A Study on the Impermeability of Nanodispersible Modified Bentonite Based on Colloidal Osmotic Pressure Mechanisms and the Adsorption of Harmful Substances

**DOI:** 10.3390/nano13121840

**Published:** 2023-06-11

**Authors:** Xi Wei, Chunyang Zhang, Depeng Gong, Mengdong Tu, Lili Wu, Wanyu Chen, Chaocan Zhang

**Affiliations:** School of Materials Science and Engineering, Wuhan University of Technology, Wuhan 430070, China; 303868@whut.edu.cn (X.W.); 13476089628@163.com (C.Z.); gdp@whut.edu.cn (D.G.); 303549@whut.edu.cn (M.T.); polym_wl@whut.edu.cn (L.W.); chenwanyu@whut.edu.cn (W.C.)

**Keywords:** bentonite, sodium polyacrylate, betaine, osmotic pressure, hydraulic conductivity, adsorption

## Abstract

With the growing demands of human beings, sanitary landfill, along with the increase in landfill depth and leachate water pressure, has put forward new and higher requirements for the impermeable layer. In particular, it is required to have a certain adsorption capacity of harmful substances from the perspective of environmental protection. Hence, the impermeability of polymer bentonite–sand mixtures (PBTS) at different water pressure and the adsorption properties of polymer bentonite (PBT) on contaminants were investigated through the modification of PBT using betaine compounded with sodium polyacrylate (SPA). It was found that the composite modification of betaine and SPA could reduce the average particle size of PBT dispersed in water (reduced to 106 nm from 201 nm) and enhance the swelling properties. As the content of SPA increased, the hydraulic conductivity of PBTS system decreases and the permeability resistance improves, while the resistance to external water pressure increases. It is proposed a concept of the potential of osmotic pressure in a constrained space to explain the impermeability mechanism of PBTS. The potential of osmotic pressure obtained by linear extrapolation of the trendline of colloidal osmotic pressure versus mass content of PBT could represent the external water pressure that the PBT resist. Additionally, the PBT also has a high adsorption capacity for both organic pollutants and heavy metal ions. The adsorption rate of PBT was up to 99.36% for phenol; up to 99.9% for methylene blue; and 99.89%, 99.9%, and 95.7% for low concentrations of Pb^2+^, Cd^2+^, and Hg^+^, respectively. This work is expected to provide strong technical support for the future development in the field of impermeability and removal of hazardous substances (organic and heavy metals).

## 1. Introduction

With deepening urbanization, industrial waste and municipal waste have increased rapidly. The current method of waste disposal is mainly sanitary landfill, an efficient way to prevent pollutants from polluting the environment [1,2]. Leachate from sanitary landfilling, however, typically contains organic contaminants and heavy metal ions, damaging both the local ecological environment and public health [3,4,5]. Organic pollutants significantly harm biological and human health, weaken the body’s defenses against illness, and result in birth abnormalities and reproductive issues [6,7,8]. Heavy metal ions could pose a major threat to ecosystems and human health even at low concentrations because of their high mobility, bioaccumulation, tremendous toxicity, and cancer-causing potential [9]. Lead and cadmium are examples of heavy metal ions that can harm the kidneys and induce analgesia [10], hepatitis, carcinogenesis [11], pulmonary fibrosis [12], and indigestion [13]. Leachate pollution of the environment can be avoided via impermeable layers [2,14]. The primary types of impermeable layers include geomembranes, geosynthetic clay liners (GCL), compacted clay liners (CCL), and bentonite–sand mixes (BS) [15]. Due to the uneven settlement of the foundation, CCL, GCL, and geomembrane may be destroyed and impermeability decreased. However, the BS is a main direction for the development of impermeable layers as it is low cost, self-healing, and has low permeability [16,17].

Bentonite is a kind of clay, with montmorillonite as the main mineral. The basic structure of the montmorillonite crystal lattice consists of silica (tetrahedron) and alumina (octahedron). Additionally, there are two primary types of bentonite: sodium bentonite and calcium bentonite [18]. As a result of the layers of montmorillonite absorbing water molecules, bentonite dictated the impermeability of BS, which showed up as macroinflation [19,20]. Bentonite may be biologically changed to increase its impermeability [21,22]. According to Salemi [17], sodium polyacrylate (SPA)-modified bentonite can reduce the hydraulic conductivity of bentonite from 1.73 × 10^−7^ m/s to 2.06 × 10^−8^ m/s. By altering bentonite with polyacrylic acid, Scalia [23] demonstrated that the hydraulic conductivity of bentonite decreased from 2 × 10^−10^ m/s to 8 × 10^−11^ m/s. Based on Ozhan [24], modified bentonite containing cationic polyacrylamide can lower hydraulic conductivity from 3.4 × 10^−10^ m/s to 5.4 × 10^−11^ m/s. Zhang [25] verified that the anionic polyacrylamide-modified bentonite reduced hydraulic conductivity from 3.91 × 10^−10^ to 2.12 × 10^−11^ m/s. A skeleton created by sand in the BS enhanced the structural durability of the impermeable layer, facilitating its long-term use [26]. The mechanism of impermeability has not yet been fully investigated. Bohnhoff [27] proposed a semi-permeable membrane mechanism that underlies bentonite resistance; Ejezie [28] contended that polyacrylamide blocks the pores between sand grains; Katsumi [16] postulated that swelling was the cause of the low hydraulic conductivity of bentonite materials; and Yu [29] claimed that polymer-modified bentonite contains minute cavity structures that lengthen the percolation path, decreasing the material’s permeability.

Bentonite has significant research value in wastewater treatment [30,31] due to its multi-layer structure, high specific surface area, and exchangeable interlayer cations, which may efficiently adsorb organic contaminants [32,33,34] and heavy metal ions [35]. However, the adsorption capacity of raw bentonite is limited, and can be increased via polymer modification [36]. Currently, the principal use of quaternary ammonium cationic surfactants is in the cation exchange modification of bentonite [37]. At a phenol concentration of 200 mg/L, He [38] discovered that modified bentonite with cetyltrimethylammonium bromide (CTAB) could achieve phenol adsorption of 10.1 mg/g. According to Meng [39], adding a quaternary ammonium surfactant to modified bentonite enhanced its ability to adsorb methylene blue from 60% to 95%. However, bentonite may become hydrophobic and have a reduced ability to adsorb metal ions if quaternary ammonium surfactants are present [40]. Liu [41,42] found that modified bentonite with amphoteric surfactants was effective at adsorbing bisphenol A, Pb^2+^, and Cd^2+^. This was probably due to the carboxyl groups (−COO−) and positively charged groups (−N^+^) that were present in the amphoteric surfactants, which make it simpler to adsorb metal ions and organic contaminants [43,44,45].

As sanitary landfills evolve, the depth of the landfill steadily deepens, the water pressure of the leachate increases and the demands on the impermeable layer are raised. This study examined the impermeability of a PBT–sand mixture (PBTS) and the composited-modification bentonite’s (PBT) adsorption capability after being treated with sodium polyacrylate and betaine. Additionally, the osmotic pressure mechanism was proposed as an impermeability mechanism. In a sand–sand constrained space, the colloidal osmotic pressure of PBTS determined the water pressure it could resist. The findings showed that the osmotic pressure of the PBT colloid increased with increasing SPA concentration, and that PBTS’s resistance to permeability and external water pressure both improved. However, the hydraulic conductivity of PBTS increased and the resistance to permeability decreased when the water pressure was greater than the osmotic pressure of the PBT colloidal. Additionally, the PBT showed effective adsorption properties for phenol, heavy metal ions, and methylene blue.

## 2. Experimental Methods

### 2.1. Materials

Natural bentonite (OB) came from India, an earth-yellow powder with a water content of 8%, particle size of 0.075 mm, and cation exchange capacity (CEC) of 67.1 mmol/100 g; the main composition is shown in Table 1. The molecular weight of sodium polyacrylate (SPA) was 5 million (500 w), 10 million (1000 w), 15 million (1500 w), 20 million (2000 w), and 25 million (2500 w), from industrial-level products. Betaine was analytically pure; the fineness of the sand was 30 mesh; and the experimental water was deionized water.

### 2.2. Preparation of Polymer-Modified Bentonite

A total of 1.5 g (3% of bentonite mass) sodium polyacrylate (SPA) dissolved in 300 mL deionized water at 70 °C and 50 g bentonite dissolved in 1000 mL deionized water were stirred for 2 h. The bentonite solution was evenly dispersed and then added to the sodium polyacrylate solution, stirred for 1h, dried at 100 °C, ground, and passed through a 200 mesh sieve, and the sample was named PB. Then, 2.5 g (5% of the bentonite mass) betaine was dissolved in 100 mL deionized water, added to the dispersed bentonite solution, and stirred for 1h, and then sodium polyacrylate solution was added, stirred for 1h, dried at 100 °C, ground, and passed through a 200 mesh sieve, and the sample was named PBT. The bentonite was modified only with betaine, and named BT. The different contents and different betaine content of bentonite were named P*_x_*BT*_y_*; *x* was the content of SPA, and *y* was the content of betaine. The different molecular weights of the SPA composite modified with betaine were named as P**_500w_**BT, P**_1000w_**BT, P**_1500w_**BT, P**_2000w_**BT, and P**_2500w_**BT.

### 2.3. Material Characterization

X-ray diffraction (XRD) was carried out using a Panaco Empyrean diffractometer from the Netherlands at 40 kV and 40 mA with Cu (λ = 1.5406), and scans were recorded between 4° and 40° in 0.1° steps at a rate of 2 °/min. The particle size distribution of the bentonite colloids was measured using a laser particle size analyzer, Master Sizer 2000, Malvern, UK. FTIR was used to test the changes in bentonite before and after modification with the Therno Nicolet Nexus smart Fourier transform infrared spectrometer, USA, with a measured spectral range of 400–4000 cm^−1^ and an accuracy of 0.01 cm^−1^. The microscopic morphology of the OB, PB, and PBT was characterized using a JSM-7500F SEM. The zetapotential of OB, PB, and PBT was tested.

### 2.4. Free Swell Index

The free swell index (FSI) test was conducted according to the ASTM D5890 standard method. Add 90mL of deionized water to a 100 mL measuring cylinder, take 2 g dried OB, PB and PBT, add about 0.1 g or so to the cylinder each time, leave an interval of at least 10 min, add water to 100 mL after all the additions. The free swell index of bentonite (mL/2 g) was measured after 24 h.

### 2.5. Hydraulic Conductivity

A total of 16 g of OB and PBT mixed with 64 g of sand, were added to 7–9 g of deionized water, and water was slowly sprayed into the sand mixture. Then, under 10 MPa pressure, they were pressed into a specimen with a height of 10 mm and a diameter of 70 mm, with a density of about 2.2 g/cm^3^. The hydraulic conductivity was measured with a TST-55 permeameter, controlling the water pressure from 20 kPa in increments of 30 kPa to 200 kPa. The hydraulic conductivity was calculated as follows:(1)$$\;k=\frac{a\times L}{2\times A\times \Delta t}\ln\left(\frac{\Delta h_{1}}{\Delta h_{2}}\right)$$
where *k* is the hydraulic conductivity (m/s), *a* is the cross-sectional area of the PBTS specimen m^2^; *L* is the height of the PBTS specimen (m); *A* is the water passage area of the sample (cm^2^); Δ*t* is the time interval(s); $\Delta h_{1}$ is the head loss across the permeameter/specimen at the start time of the permeation trial (m); and $\Delta h_{2}$ is the head loss across the permeameter/specimen at the end time of the permeation trial (m).

### 2.6. Colloidal Osmotic Pressure

PBT and OB were dissolved in 100 mL of deionized water to prepare colloidal solutions with concentrations of 5, 15, 25, 35, 45, and 55 g/L. The osmotic molar concentrations of bentonite colloids were measured using a German GONOTEC freezing point osmotic pressure meter, Osmomat030 3000.

### 2.7. Adsorption Experiments

#### 2.7.1. Phenol Adsorption Experiments

OB and PBT were dissolved in 100 mL of phenol solution with a solution concentration of 50, 100, 200, 400, 600, and 800 mg/L, shaken at 150 r/min for 3 h. The supernatant was removed and centrifuged at 7500 r/min for 15 min and passed through a 0.45 μm filter membrane, and the absorbance of the filtrate at 271 nm was measured.

#### 2.7.2. Methylene Blue Adsorption Experiments

OB and PBT were dissolved in 50 mL of methylene blue solution with a solution concentration of 200 mg/L, respectively, shaken at 25 °C at 150 r/min for 3 h, the supernatant taken, and the absorbance measured at 664 nm using a UV spectrophotometer.

#### 2.7.3. Heavy Metal Ion Adsorption Experiments

OB and PBT were dissolved in 100 mL of Pb^2+^, Cd^2+^, Hg^+^ ion solution with a solution concentration of 200 mg/L, shaken at 25 °C with 150 r/min for 3 h, the supernatant taken, centrifuged at 7500 r/min for 15 min, passed through a 0.45 μm filter membrane, and the metal ions in the filtrate measured using flame method atomic absorption spectroscopy.

## 3. Results and Discussions

### 3.1. Characterization

XRD was used to analyze the structures of OB, PB, PBT, and BT. The results are shown in Figure 1a. The d_001_ of PBT was reduced from 1.451 nm to 1.223 nm as compared to OB, which contributed to the quaternary amino via ion exchange in betaine. The bentonite layer distance of d_001_ was not significantly affected by a betaine content above 5% (Appendix A). The particle sizes of OB, PB, and PBT dispersed in water are shown in Figure 1b. PBT displayed nanodispersibility after modification, with the particle size dropping from 203 nm for OB to 106 nm, due to the intercalation of betaine. The IR of bentonite is shown in Figure 1c, where the main spectral bands of bentonite are 3623 cm^−1^, attributed to the stretching vibration of montmorillonite structural hydroxyl −OH, 3436 cm^−1^, and 1636 cm^−1^, mainly due to the stretching vibration of an interlayer water molecule −OH and bending vibration of −OH, 1421 cm^−1^ attributed to the stretching vibration and bending vibration of C-H, 1034 cm^−1^ and 797 cm^−1^ attributed to the stretching vibration and bending vibration of Si-O, and 876 cm^−1^ attributed to the bending vibration of montmorillonite structural hydroxyl −OH. In particular, the peak at 1564 cm^−1^ was the stretching vibration of acrylate (−COO−) on both PB and PBT, and the peak at 1339 cm^−1^ was the stretching vibration of C-N on PBT.

### 3.2. Free Swell Index

The higher the FSI of bentonite, the lower its hydraulic conductivity [22]. The FSI values of OB, P**_3%_**B (3%SPA) and P**_3%_**BT**_5%_** (3% SPA and 5% betaine) are shown in Figure 2a. As the SPA molecular weight increased, the FSI gradually rose, and betaine caused the FSI to rise even higher. The FSI of PBT**_5%_** (5% betaine) is shown in Figure 2b; the higher the content of SPA, the more significantly the FSI increased. The FSI of P**_3%&2000w_**BT (3% SPA (2000 w, molecular weight)) is shown in Figure 2c, where the FSI increased with the increased betaine content, with no further increase beyond 5%. Figure 2d shows that following the SPA treatment, the zetapotential of PBT increased and its stability was improved; the stability increased with increasing SPA content. The zetapotential continued to rise following SPA and betaine compound modification, and stability improved with increasing betaine content.

### 3.3. Hydraulic Conductivity

The impermeability of PBTS combined with PBT and sand in various ratios was examined. Based on the results of the FSI experiment, PBT was made by modifying 5% betaine and SPA with various molecular weights and contents. The hydraulic conductivity of P**_2000w_**BT**_5%_**S (SPA (2000 w, molecular weight) and 5% betaine) with a 20:80 PBT–sand ratio at 20–200 kPa water pressure is shown in Figure 3b. The result indicated the P**_2000w_**BT**_5%_**S with high SPA content had a lower hydraulic conductivity and better impermeability. With the rising water pressure, the hydraulic conductivity of P**_2000w_**BT**_5%_**S with varied SPA contents first fell and then rose. The critical water pressure P_c_ was defined as the water pressure at the lowest value of hydraulic conductivity, as shown in Table 2, which rose as the content of SPA increased.

Generally, the hydraulic conductivity of PBTS was influenced by the PBT–sand ratio. As shown in Figure 3c, the hydraulic conductivity of P**_5%&2000w_**BT**_5%_**S (5% SPA (2000 w, molecular weight) and 5% betaine) decreased first and then increased when the PBT–sand ratio increased, coinciding with earlier studies [46]. The P**_c_** for P**_5%&2000w_**BT**_5%_**S, with the 15:85 PBT–sand ratio lower than the other PBT–sand ratios of PBTS, due to the low PBT content of the system, did not have a higher colloidal osmotic pressure and had relatively poor impermeability. Additionally, the hydraulic conductivity of P**_5%_**BT**_5%_**S with different SPA molecular weights and 20:80 PBT–sand ratios is shown in Figure 3d, and decreased when the molecular weight increased; at 170 kPa, the lowest hydraulic conductivity of P**_2500w_**BT**_5%_**S was 4.62 × 10^−12^ m/s.

### 3.4. Colloidal Osmotic Pressure Mechanism

Our experimental results were not well explained by current impermeability mechanisms, so the colloidal osmotic pressure mechanism was proposed. Figure 4a depicts the osmotic pressure of the PBT colloidal solution with various SPA concentrations. The figure showed that the greater the SPA concentration of PBT, the higher the osmotic pressure, which was consistent with the experimental findings. Moreover, the osmotic pressure tended to flatten out and alter almost linearly as the mass concentration increased. Obviously, the impermeable PBT had a highly concentrated colloidal system, in which colloidal osmotic pressure was hardly measured. Thus, the curve of osmotic pressure with high concentration was extrapolated to 100% by the PBT mass content (seen in Figure 4d). The intercept value gained from extrapolating, shown in Table 2, was called the potential of osmotic pressure (**π_µ_**). The **π_µ_** could represent the chemical potential of PBT, which was an ability to form osmotic pressure. The test value P_c_ basically coincided with the **π_µ_**, indicating that the ability of PBTS to resist external water pressure depended on the colloidal osmotic pressure in a constrained space. Thus, the colloidal osmotic pressure mechanism was proposed to explain the impermeability of PBTS (shown in Figure 5). When the PBTS was compacted, the sand particles overlapped each other to form a sand–sand constrained space which the PBT filled. When water began to penetrate, the PBT absorbed it and expanded, gradually creating a highly concentrated colloidal system with a comparatively stable colloidal osmotic pressure. Due to the PBT not being able to absorb more water and expand in a constrained space, it was able to resist water penetration. When the external water pressure exceeded the colloid osmotic pressure, the PBT colloid loss and impermeability of PBTS were reduced.

### 3.5. Adsorption Properties

Domestic waste, industrial wastewater, and mine tailings usually contain organic pollutants and heavy metal ions, but the nanodispersible PBT may have great adsorption performance for harmful substances. In this paper, the organic pollutants and heavy metal ions adsorption performance of OB and PBT were investigated, and the adsorption processes of OB and PBT were described by Langmuir and Freundlich’s isothermal adsorption models and adsorption kinetics [47,48].

#### 3.5.1. Phenol Adsorption

The adsorption capacity of P**_3%&2000w_**BT**_5%_** (3% SPA (2000 w, molecular weight) and 5% betaine) for different concentrations of phenol is shown in Figure 6a; it increased as the phenol concentration rose, and reached 150 mg/g. Compared to OB, the phenol adsorption capacity of PBT improved by 289% at a 50 mg/L concentration. It was higher than the 120.4 mg/g reported in previous research for bentonite modified by dodecyldimethyl betaine [49]. The phenol adsorption rate of P**_3%_**BT**_5%_** with different molecular weights of SPA is shown in Figure 6b, and it could reach more than 90% and up to 99%. When the SPA content increased, as shown in Figure 6c, the phenol adsorption increased up to 111 mg/g. The phenol adsorption increased up to 111 mg/g as the SPA concentration rose, as illustrated in Figure 6c. Figure 6d depicted the P**_3%&2000w_**BT’s phenol adsorption capacity with various betaine contents; as the betaine content grew, the phenol adsorption capacity increased up to 102 mg/g. Because of its high surface tension, phenols with low surface tension could not be effectively adsorbed onto OB. After modification, PBT exhibited a higher affinity for phenol. As shown in Figure 7, the phenol adsorption processes of OB and PBT were more consistent with the Freundlich model and with the pseudo-second-order kinetics (Appendix A).

#### 3.5.2. Methylene Blue Adsorption

The adsorption performance of PBT on organic pollutants using methylene blue as a simulant was investigated. With increasing molecular weight, as shown in Figure 8a,b, the methylene blue adsorption capacity and adsorption rate of P**_3%_**BT**_5%_** (3% SPA and 5% betaine) both increased from a maximum of 335 mg/g in OB to 464 mg/g, an increase of 38.5%, and from 91.3% of OB to 99.9%, respectively. Figure 8c depicts the methylene blue adsorption capability of P**_2000w_**BT**_5%_** with varying SPA content, which increased with increasing SPA content. As demonstrated in Figure 8d, the methylene blue adsorption capability of P**_2000w_**BT**_5%_** increased as the betaine concentration increased. The adsorption process of PBT was via pseudo-second-order kinetics (Appendix A),Appendix A.

#### 3.5.3. Heavy Metal Ion Adsorption

As shown in Figure 9a, the Pb^2+^ adsorption of P**_2000w_**BT**_5%_** (with 5% betaine and different contents SPA (2000 w, molecular weight)) rose from 128 mg/g in OB to 151 mg/g with increasing SPA concentration. Additionally, this was similar to the 150 mg/g reported in previous research for bentonite modified by sodium polyacrylate [50]. With betaine content increasing, as shown in Figure 9b, the adsorption capacity of P**_3%&2000w_**BT increased. The adsorption process of P**_3%_**BT**_5%_** was pseudo-second-order kinetic (Appendix A),Appendix A). In fact, the content of heavy metal ions in domestic waste was low, and the adsorption rate of PBT on heavy metal ions at low concentrations (20 mg/L) could be investigated. Figure 9c depicted the Pb^2+^, Cd^2+^, and Hg^+^ adsorption rates of OB and PBT with various SPA molecular weights. When compared to OB, all of the Pb^2+^ adsorption rates of PBT with different SPA molecular weights increased by 12%. The Cd^2+^ adsorption rates of P**_3%_**BT**_5%_** at various molecular weights were comparable to those of Pb^2+^, which increased by 10% and exceeded 99%; however, the Hg^+^ adsorption rates of P**_3%_**BT**_5%_** at various molecular weights of SPA increased with increasing molecular weight, reaching a maximum of 95.7% and an increase of 35% over OB. The heavy metal ions adsorption capacity of PBT was enhanced by electrostatic adsorption, increased adsorption sites and complexation of the polymer with metal ions, and basically the complete adsorption of heavy metal ions at low concentrations, which could effectively remove trace contents of heavy metal ions from the waste filtrate.

PBT had good adsorption capabilities for both organic contaminants and heavy metal ions, contrasted with quaternary ammonium salt modifications [43]. This was primarily due to the complex modification of the amphoteric surfactant betaine with SPA, which improved the hydrophilicity and dispersibility of the bentonite while increasing the number of adsorption sites.

## 4. Conclusions

In this study, bentonite was treated with SPA and betaine to reduce the particle size from 203 nm to 106 nm and achieve nanoscale dispersion. The swelling performance of PBT was improved, and the zetapotential indicated that the stability of PBT was improved. The PBTS impermeability was investigated from the standpoint of osmotic pressure, and the colloidal osmotic pressure mechanism was proposed to explain the PBTS impermeability process. When the colloidal osmotic pressure of PBT increased, its permeability coefficient decreased, its impermeability improved, and its resistance to external water pressure increased. The permeability coefficient of PBTS could be lowered to 4.62 × 10^−12^ m/s after modification. PBT was also effective in absorbing organic pollutants and heavy metal ions. The phenol adsorption capacity and adsorption rate of PBT went up to 150 mg/g and 99.36%, respectively; the methylene blue adsorption capacity and adsorption rate of PBT went up to 464 mg/g and 99.91%, respectively; the Pb^2+^ adsorption capacity went up to 151 mg/g; and the Pb^2+^, Cd^2+^, Hg^+^ adsorption rate of PBT in low concentrations went up to 99.89%, 99.9%, and 95.7%, respectively. Overall, this work could provide strong technical support for future developments in the field of impermeability and the removal of hazardous substances (organic and heavy metals).

## Figures and Tables

**Figure 1 nanomaterials-13-01840-f001:**
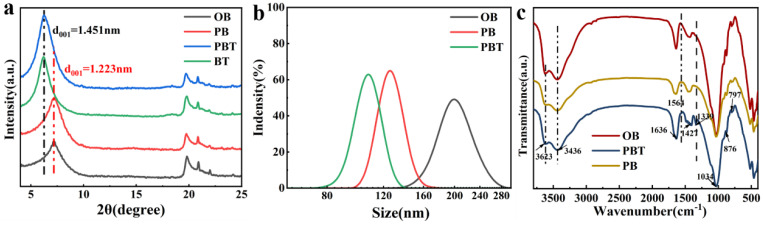
(**a**) XRD patterns of OB, PB, PBT, and BT; (**b**) PSD patterns of OB, PB, and PBT; (**c**) IR patterns of OB, PB, and PBT.

**Figure 2 nanomaterials-13-01840-f002:**
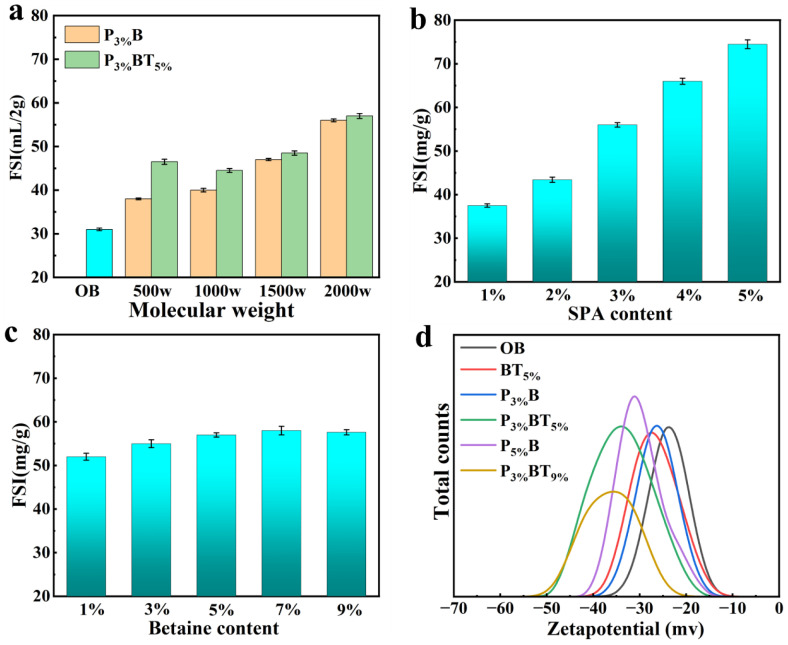
(**a**) The FSI of OB, P**_3%_**B, and P**_3%_**BT**_5%_** with different molecular weights of SPA; (**b**) the FSI of PBT**_5%_** with different SPA (2000 w) content; (**c**) the FSI of P**_3%&2000w_**BT with different betaine content; and (**d**) the zetapotential of OB and PB with different SPA contents and PBT with different SPA and betaine contents.

**Figure 3 nanomaterials-13-01840-f003:**
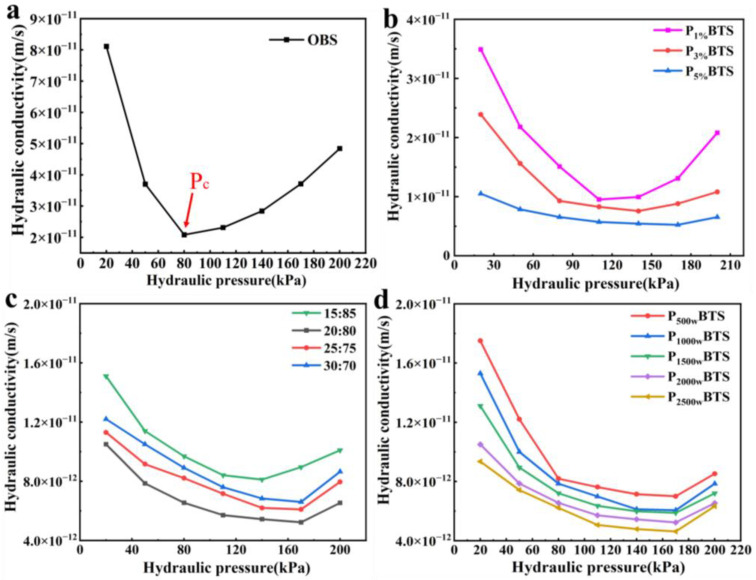
(**a**) Hydraulic conductivity of OBS; (**b**) hydraulic conductivity of P**_2000w_**BT**_5%_**S with different SPA contents; (**c**) the hydraulic conductivity with different PBT–sand ratios for P**_5%&2000_**_w_BT**_5%_**S; and (**d**) the hydraulic conductivity with different SPA molecular weights for P**_5%_**BT**_5%_**S.

**Figure 4 nanomaterials-13-01840-f004:**
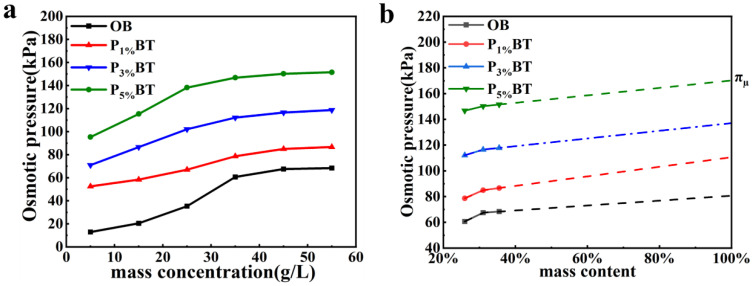
(**a**) Osmotic pressure of OB, P**_1%_**BT, P**_3%_**BT, and P**_5%_**BT colloidal at different mass concentrations; (**b**) linear extrapolation of osmotic pressure based on bentonite mass content.

**Figure 5 nanomaterials-13-01840-f005:**
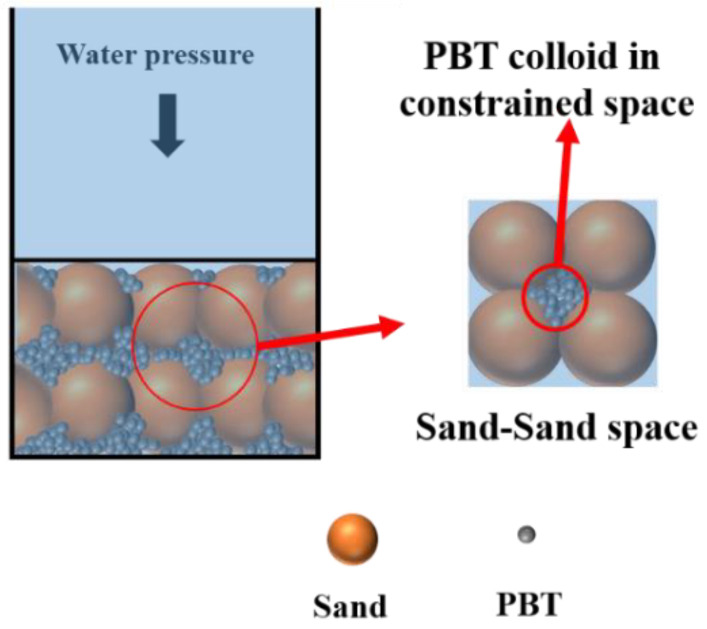
Colloidal osmotic pressure mechanism of PBTS.

**Figure 6 nanomaterials-13-01840-f006:**
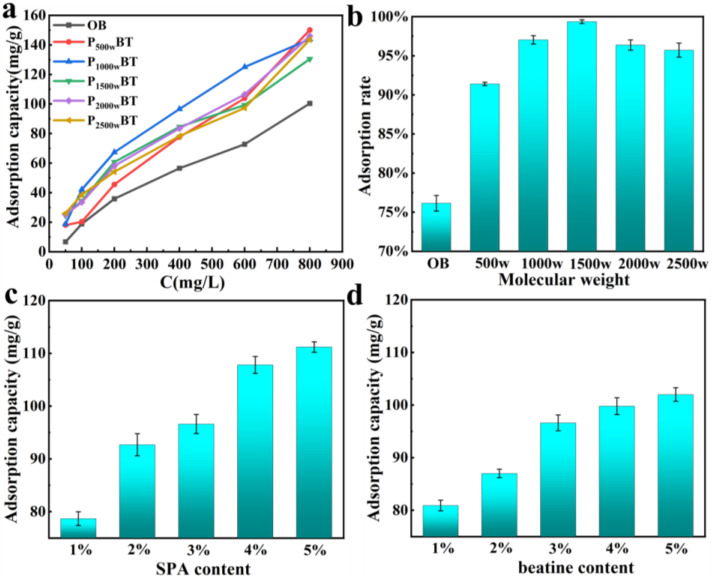
(**a**) The phenol adsorption capacity of P**_3%&2000w_**BT**_5%_** with different phenol concentrations; (**b**) the phenol absorption capacity of P**_3%_**BT**_5%_** with different SPA molecular weights; (**c**) the phenol adsorption of P**_2000w_**BT**_5%_** with different SPA contents; and (**d**) the phenol adsorption of P**_3%&2000w_**BT with different betaine contents.

**Figure 7 nanomaterials-13-01840-f007:**
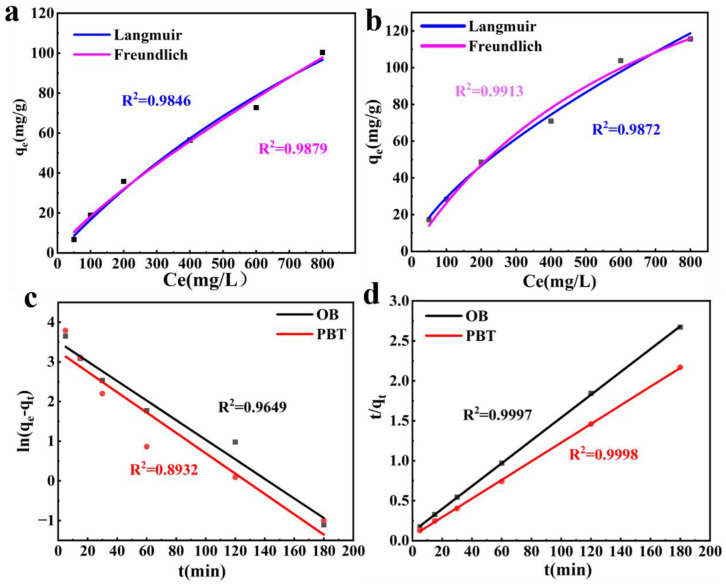
(**a**) OB isothermal adsorption model; (**b**) PBT isothermal adsorption model; (**c**) the pseudo-first-order kinetics of OB and PBT; and (**d**) the pseudo-second-order kinetics of OB and PBT.

**Figure 8 nanomaterials-13-01840-f008:**
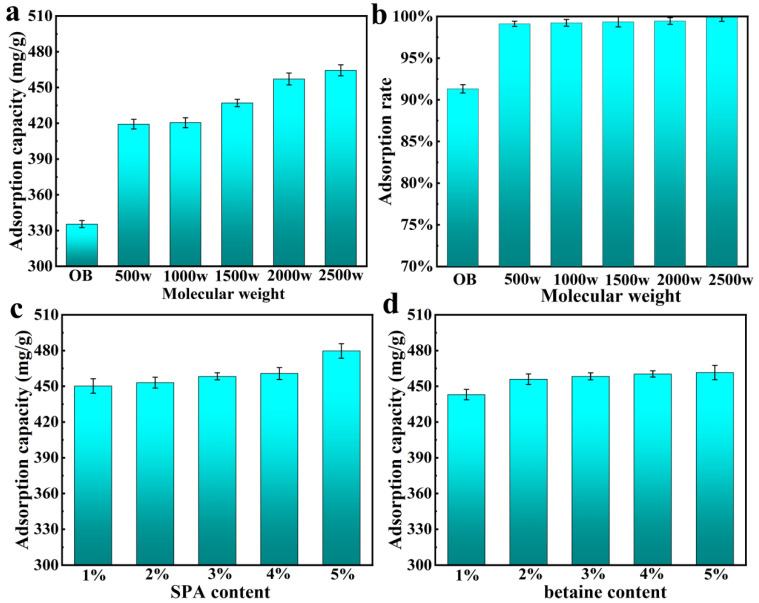
(**a**) The methylene blue adsorption capacity of P**_3%_**BT**_5%_** with different SPA molecular weights; (**b**) the methylene blue adsorption rate of P**_3%_**BT**_5%_** with different SPA molecular weights; (**c**) the methylene blue adsorption capacity of P**_2000 w_**BT**_5%_** with different SPA contents; and (**d**) the methylene blue adsorption capacity of P**_2000w_**BT**_5%_** with different betaine contents.

**Figure 9 nanomaterials-13-01840-f009:**
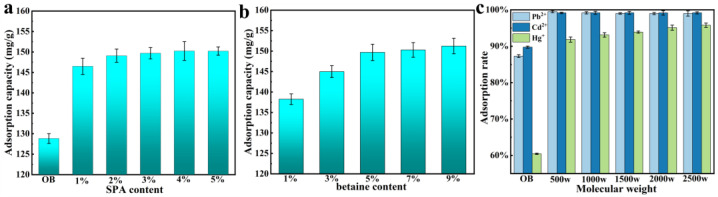
(**a**) The Pb^2+^ adsorption capacity of P**_2000w_**BT**_5%_** with different SPA contents; (**b**) the Pb^2+^ adsorption capacity of P**_3%&2000w_**BT with different betaine contents; and (**c**) the Pb^2+^, Cd^2+^ and Hg^+^ adsorption rates of OB and P**_3%_**BT**_5%_** with different SPA molecular weights.

**Table 1 nanomaterials-13-01840-t001:** Main components of bentonite (%mass).

Montmorillonite	Kaolinite	Quartz	Halloysite
63%	19%	14%	3%

**Table 2 nanomaterials-13-01840-t002:** P_c,_
**π_µ_** with various SPA contents (kPa).

	0%	1%	3%	5%
P_c_	80	110	140	170
**π_µ_**	80.6	110.5	137	170.2

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
