# Peer review of "A Study on the Impermeability of Nanodispersible Modified Bentonite Based on Colloidal Osmotic Pressure Mechanisms and the Adsorption of Harmful Substances"

_nanomaterials, 2023, doi:10.3390/nano13121840_

Round 1
Reviewer 1 Report
Dear Authors,
The manuscript presents the study of the impermeability of modified bentonite based on colloidal osmotic pressure mechanism. The Abstract contains information relevant to the study. In general the manuscript is well structured.
The Introduction could be improved. In the Introduction, at least a briefly definition of what bentonite is (and how many types of bentonite are known and so on) should be added. This will make a better overview and will be useful for the readers, especially if they are not directly involved in the fields of bentonite, but they are interested for such studies too, as the one described in this manuscript.
All the abbreviation are explained when they first appear. The Results and discussion are sustained by the data and graphs. As about the swelling properties of bentonite, in addition to free swell index, it will be of great interest if some results of electrokinetic potential (i.e. zeta potential or surface potential) curves could be shown as well. Or some calculations of it. The zeta potential characterizes among others, the swelling process.
The IR of bentonite showed only the vibration of COO- and C-N? Are those the only IR results? If there are more relevant IR results, please add them.
The adsorption could not be discussed also in terms of surface tension or surface behavior?
The Conclusions are in agreement with the described results.
The manuscript needs a minor revision before to be accepted for publication.
Best regards!

The quality of English is good. The are only few minor mistakes. Please check carefully the whole manuscript and improve it, where it is the case.
Author Response
We sincerely thank you all for spending precious time and efforts in examining this manuscript and greatly appreciate your insightful comments to make the paper better. The manuscript has been carefully revised, improved and verified to address the questions raised by the reviewers, and revisions have been marked with yellow for clarification. Below we list responses in the sequence in which they were raised in the referee’s report.
Comments and Suggestions for Authors.
Point 1. The Introduction could be improved. In the Introduction, at least a briefly definition of what bentonite is (and how many types of bentonite are known and so on) should be added. This will make a better overview and will be useful for the readers, especially if they are not directly involved in the fields of bentonite, but they are interested for such studies too, as the one described in this manuscript.
Response 1: We are really grateful for your valuable comments and suggestions. Bentonite is a kind of clay with montmorillonite as the main mineral. The basic structure of montmorillonite crystal lattice consists of silica (tetrahedron) and alumina (octahedron). And there were two primary types of bentonite: sodium bentonite and calcium bentonite. Please see page 1, line 21-24.
Point 2. All the abbreviation are explained when they first appear. The Results and discussion are sustained by the data and graphs. As about the swelling properties of bentonite, in addition to free swell index, it will be of great interest if some results of electrokinetic potential (i.e. zeta potential or surface potential) curves could be shown as well. Or some calculations of it. The zeta potential characterizes among others, the swelling process.
Response 2: We are really grateful for your valuable comments and suggestions. We have added the zeta potential to characterize the swelling of bentonite. After SPA modification, the zetapotential of PBT increased and its stability was improved; the stability increased with increasing SPA content. The zetapotential continued to rise following SPA and betaine compound modification, and stability improved with increasing betaine content. Please see page8, line 200-204.
Point 3. The IR of bentonite showed only the vibration of COO- and C-N? Are those the only IR results? If there are more relevant IR results, please add them.
Response 3: We are really grateful for your valuable comments and suggestions. We have added the other IR results. the main spectral bands of bentonite are: 3623 cm-1 attributed to the stretching vibration of montmorillonite structural hydroxyl -OH, 3436 cm-1, and 1636 cm-1 mainly due to the stretching vibration of an interlayer water molecule -OH and bending vibration of -OH, 1421 cm-1 attributed to the stretching vibration and bending vibration of C-H, 1034 cm-1 and 797 cm-1 attributed to the stretching vibration and bending vibration of Si-O, and 876 cm-1 attribute to the bending vibration of montmorillonite structural hydroxyl -OH. Specially, the peak at 1564 cm-1 was stretching vibration of acrylate (-COO-) on both PB and PBT, and peak at 1339 cm-1 was stretching vibration of C-N on PBT. Please see page7, line 176-183.
Point 4.The adsorption could not be discussed also in terms of surface tension or surface behavior?
Response 4: We are really grateful for your valuable comments and suggestions. Because of its high surface tension, phenols with low surface tension cannot be effectively adsorbed onto OB. After modification, PBT exhibited a higher affinity for phenol. Please see page12, line 289-291.

Reviewer 2 Report
In the paper entitled "Study on the impermeably of nanodispersible modified bentonite based on colloidal osmotic pressure mechanism and adsorption of harmful substances", the authors present a series of experimental results showing that modifying bentonite improves its absorption and reduces its transport properties. The experiments are well planned and executed. The conclusions drawn from the studies are logical. However, I suggest rethinking and taking a stand on the following point. According to the authors, eq. (1) represents the hydraulic conductivity (k) expressed i m/s. In hydrodynamics, volume flux (Jv = dV/Sdt) is expressed in m/s. From Poiseuille's equation it follows that Jv =r2dP/8lf =khdP. From here it follows that kh = Jv/dP and its unit is m3/Ns. In this case, how does k relate to kh.
Author Response
We sincerely thank you all for spending precious time and efforts in examining this manuscript and greatly appreciate your insightful comments to make the paper better. The manuscript has been carefully revised, improved and verified to address the questions raised by the reviewers, and revisions have been marked with yellow for clarification. Below we list responses in the sequence in which they were raised in the referee’s report.
Reviewer: 2
Point 1. In the paper entitled "Study on the impermeably of nanodispersible modified bentonite based on colloidal osmotic pressure mechanism and adsorption of harmful substances", the authors present a series of experimental results showing that modifying bentonite improves its absorption and reduces its transport properties. The experiments are well planned and executed. The conclusions drawn from the studies are logical. However, I suggest rethinking and taking a stand on the following point. According to the authors, eq. (1) represents the hydraulic conductivity (k) expressed i m/s. In hydrodynamics, volume flux (Jv = dV/Sdt) is expressed in m/s. From Poiseuille's equation it follows that Jv =r2dP/8lf =khdP. From here it follows that kh = Jv/dP and its unit is m3/Ns. In this case, how does k relate to kh.
Response 1: We are really grateful for your valuable comments and suggestions。 There is a close relationship between the flow coefficient Kh and the permeability coefficient k. Kh describes the liquid flow rate per unit area through the medium, while the permeability coefficient k describes the resistance of the medium to liquid flow. The exact relationship between the infiltration coefficient (k) and the flow coefficient (Kh) might vary depending on the qualities of the soil or rock involved, the flow circumstances, and the theoretical model used. Therefore, it is impossible to find a stable formula that just connects these two coefficients. Depending on the unique engineering or groundwater flow issues, the relationship between them needs to be established or validated using experimental data.

Reviewer 3 Report
The manuscript titled « Study on the impermeability of nanodispersible modified bentonite based on colloidal osmotic pressure mechanism and adsorption of harmful substances” submitted in the journal of Nanomaterials is not suitable for publication in its present form.
1) The English is very poor and need extensive revision. it seems that the authors use google translate. (Example: the adsorption performance of composited-modification bentonite (PBT), modified with betaine and sodium polyacrylate, and the impermeability of PBT-sand mixture (PBTS) were investigated).
2) The abstract should be rewritten and should contains the most interesting results obtained in this work.
3) The swell index of bentonite is mL/2 g, why 2g ? it will be better if the unit is mL/g
4) The results obtained from the XRD showed decreases of the particles size and the layer distance is affected, so how to confirm the improvement of the swelling properties?
5) The adsorption properties of the materials should be compared with other materials to confirm the originality of this work.
6) The conclusion should be reformulated according the obtained results and should be coherent with the abstract.
English language is poor and could be cheked
Author Response
We sincerely thank you all for spending precious time and efforts in examining this manuscript and greatly appreciate your insightful comments to make the paper better. The manuscript has been carefully revised, improved and verified to address the questions raised by the reviewers, and revisions have been marked with yellow for clarification. Below we list responses in the sequence in which they were raised in the referee’s report.
Reviewer: 3
Point 1. The English is very poor and need extensive revision. it seems that the authors use google translate. (Example: the adsorption performance of composited-modification bentonite (PBT), modified with betaine and sodium polyacrylate, and the impermeability of PBT-sand mixture (PBTS) were investigated).
Response 1: We are really grateful for your valuable comments and suggestions. We have already improvement the English,and the proposed areas were carefully revised. Please revisit this paper and see the abstract.
Point 2. The abstract should be rewritten and should contains the most interesting results obtained in this work.
Response 2: We are really grateful for your valuable comments and suggestions. We have rewritten the abstract and highlighted the most interesting results and please see the page 1.
Point 3. The swell index of bentonite is mL/2 g, why 2g? It will be better if the unit is mL/g
Response 3: We are really grateful for your valuable comments and suggestions. The free swell index test was performed on bentonite according to the American Society for Testing and Materials’ method ASTM D5890 and the mL/2 g is a prescribed unit.
Point 4. The results obtained from the XRD showed decreases of the particles size and the layer distance is affected, so how to confirm the improvement of the swelling properties?
Response 4: We are really grateful for your valuable comments and suggestions. The swelling index, which is correlated with bentonite particle size and dispersion in water, measures the bentonite's capacity to absorb water and swell in water under static conditions. SPA and betaine compound modified bentonite, bentonite layer spacing increases, particle size lowers, water dispersion increases, and specific surface area increases all can improve the swelling properties.
Point 5. The adsorption properties of the materials should be compared with other materials to confirm the originality of this work.
Response 5: We are really grateful for your valuable comments and suggestions. We have added comparative information on the adsorption properties of the bentonite modified by dodecyldimethyl betaine and the bentonite modified by sodium polyacrylate. Please see page 11, line 280-281 and page 14, line 326-327.
Point 6. The conclusion should be reformulated according the obtained results and should be coherent with the abstract.
Response 6: We are really grateful for your valuable comments and suggestions. We have rewritten the conclusion, reformulated the obtained results, and made them coherent with the abstract. Please see page 14-15, line 347-360.

Round 2
Reviewer 2 Report
Dear authors,
so it's just a matter of nomenclature. Ok
Reviewer 3 Report
Now the revised can be accepted as is in MDPI-Nanomaterials Journal.